Chemical analysis of callus extracts from toxic and non-toxic varieties of Jatropha curcas L.

Leyva-Padrón Gerardo
Vanegas-Espinoza Pablo Emilio
Evangelista-Lozano Silvia
Del Villar-Martínez Alma Angélica
Bazaldúa Crescencio cbazaldua@ipn.mx
Centro de Desarrollo de Productos Bióticos, Instituto Politécnico Nacional , Yautepec , Morelos , México
Urban Pawel
Electronic publication date: 2020 Nov 11
Publication date: 2020
Volume: 8
Electronic Location ID: e10172
Received 2020 Feb 14; Accepted 2020 Sep 22
Copyright: ©2020 Leyva-Padrón et al.
Copyright year: 2020
Copyright holder: Leyva-Padrón et al.
License: This is an open access article distributed under the terms of the Creative Commons Attribution License, which permits unrestricted use, distribution, reproduction and adaptation in any medium and for any purpose provided that it is properly attributed. For attribution, the original author(s), title, publication source (PeerJ) and either DOI or URL of the article must be cited.
License URL: https://creativecommons.org/licenses/by/4.0/

Keywords: Jatropha curcas, micrOTOF Q-II, Callus, Phorbol esters, Glycosilated flavonoids

Funding: Instituto Politécnico Nacional IPN/COFAA/SIP-20195486 This work was supported by Instituto Politécnico Nacional (IPN/COFAA/SIP-20195486). The funders had no role in study design, data collection and analysis, decision to publish, or preparation of the manuscript.

==============================
Jatropha curcas L. belongs to Euphorbiaceae family, and it synthesizes flavonoid and diterpene compounds that have showed antioxidant, anti-inflammatory, anticancer, antiviral, antimicrobial, antifungal and insecticide activity. Seeds of this plant accumulate phorbol esters, which are tigliane type diterpenes, reported as toxic and, depending on its concentration, toxic and non-toxic varieties has been identified. The aim of this work was to characterize the chemical profile of the extracts from seeds, leaves and callus of both varieties (toxic and non-toxic) of Jatropha curcas, to verify the presence of important compounds in dedifferentiated cells and consider the possibility of using these cultures for the massive production of metabolites. Callus induction was obtained using NAA (1.5 mg L−1) and BAP (1.5 mg L−1) after 21 d for both varieties. Thin layer chromatography analysis showed differences in compounds accumulation in callus from non-toxic variety throughout the time of culture, diterpenes showed an increase along the time, in contrast with flavonoids which decreased. Based on the results obtained through microQTOF-QII spectrometer it is suggested a higher accumulation of phorbol esters, derived from 12-deoxy-16-hydroxy-phorbol (m/z 365 [M+H]+), in callus of 38 d than those of 14 d culture, from both varieties. Unlike flavonoids accumulation, the MS chromatograms analysis allowed to suggest lower accumulation of flavonoids as the culture time progresses, in callus from both varieties. The presence of six glycosylated flavonoids is also suggested in leaf and callus extracts derived from both varieties (toxic and non-toxic), including: apigenin 6-C-α-L-arabinopyranosyl-8-C-β-D-xylopyranoside (m/z 535 [M+H]+), apigenin 4′-O-rhamnoside (m/z 417 [M+H]+), vitexin (m/z 433 [M+H]+), vitexin 4′-O-glucoside-2″-O-rhamnoside (m/z 741 [M+H]+), vicenin-2 (m/z 595 [M+H]+), and vicenin-2,6″-O-glucoside (m/z 757 [M+H]+).

Introduction

Jatropha curcas L. (Euphorbiaceae) is a multipurpose plant native to Mesoamerica, and it is important because of its usefulness as raw material in biofuels production (Salvador-Figueroa et al., 2015) as well as in veterinary and human traditional medicine (Zhang et al., 2017). Several compounds with different biological activities have been isolated from different species of Jatropha (Ferreira-Rodrigues et al., 2016; Katagi et al., 2016). The identification of biologically active compounds extracted from different organs of this plant has been reported (Prasad, Izam & Khan, 2012; Sharma, Dhamija & Parashar, 2012). Isolated compounds or whole plant extracts have been studied because of their potential pharmacological activity (Cocan et al., 2018). Biological effects of J. curcas include antibacterial (Rampadarath, Puchooa & Jeewon, 2016), cytotoxic (Katagi et al., 2016), anti-inflammatory (Salim et al., 2018), and antifungal effects (Abdelgader, Suleiman & Ali, 2019; Srinivasan, Palanisamy & Mulpuri, 2019). Most research on J. curcas have been done with toxic varieties; toxicity is referred to phorbol esters content in seeds.

In Mexico, Brazil and India, non-toxic varieties of this species have been identified with very low or non-detectable levels of phorbol esters (PEs) in seeds (Laviola et al., 2010; Martínez-Herrera, Chel-Guerrero & Martínez-Ayala, 2004; Kumar, Anand & Reddy, 2011). PEs are known as Jatropha factors because each one of them has the same nucleus diterpene moiety, namely, 12-deoxy-16-hydroxy-phorbol (DHP) which is coupled to unstables intramolecular diterpenes (named C1–C6 factors) (Hirota et al., 2017).

Plants are the most successful source of chemical compounds, and their potential mode of action makes them an alternative phytomedicinal drug, since several natural products have shown benefits against human diseases (Aye et al., 2019). Several compounds are tissue-specific accumulated, and are usually structurally complex (Armaly et al., 2015). Therefore, it is necessary the use of chemical analysis techniques to isolate and identify the extracted plant metabolites (Hernandez & Sarlah, 2019). There are a few cases where the use of the plant cell culture of Jatropha curcas has allowed the production of bioactive compounds (Alvero-Bascos & Ungson, 2012; Mahalakshmi, Eganathan & Parida, 2013; Nassar et al., 2013; Zaragoza-Martínez et al., 2016); the study of the culture at different stages of toxic and non-toxic varieties generates the opportunity to design biotechnological models for production of bioactive compounds i.e., terpenoids, alkaloids, flavonoids (Abdelgadir & Van Staden, 2013; Sabandar et al., 2013), providing opportunities for new drugs discovery.

Secondary metabolites are generally in complex matrices at very low concentrations in plant organs, and lower in dedifferentiated cells. These compounds have a wide range of polarities; therefore, it is necessary the use of solvents with different polarity to obtain the extracts (Chemat et al., 2019). The aim of this work was to characterize the chemical profile of the extracts from callus of both varieties (toxic and non-toxic) of Jatropha curcas, through the cell culture, to verify the presence of important compounds in dedifferentiated cells and consider the possibility of using these cultures for the massive production of bioactive compounds.

Materials & Methods

Plant material

Seeds and young leaves of Jatropha curcas were collected. Non-toxic variety samples from Centro de Desarrollo de Productos Bióticos-IPN, Yautepec, Morelos, México (18°53′09″N, 99°03′38″W). The toxic variety samples were collected from Campo Experimental Zacatepec, Instituto Nacional de Investigaciones Agrícolas y Pecuarias (INIFAP), Zacatepec, Morelos, México (18°39′23″N, 99°11′28″W).

To induce cell dedifferentiation, two different explants were surface-sterilized according to Vanegas-Espinoza et al. (2002). Leaf blade of approximately 0.25 cm2 and petiole of approximately three mm in length were cultured in MS medium (Murashige & Skoog, 1962) supplemented with sucrose (30 g L−1), phytagel (3 g L−1) (Sigma-Aldrich®). Since there are no reports of the induction of dedifferentiated cells in the varieties analyzed in this study, the combinations of three concentrations (0.0, 1.5 and 3.0 mg L−1) of naphthaleneacetic acid (NAA) and 6-benzyl-aminopurine (BAP) were evaluated according to Verma (2013), pH was adjusted to 5.7, media were sterilized at 121 °C for 15 min. Ten explants per Petri dish with 3 repetitions per treatment were incubated at 25 ± 2 °C, photoperiod of 16 h light/8 h darkness for 35 d (Kumar et al., 2015). Explants dedifferentiation was recorded every seven days using a stereoscopic microscope (Nikon, model SMZ 1500, Japan). In order to observe differences in accumulation of compounds during callus development, completely dedifferentiated cells were cultured under the above described conditions for 38 d, samples were taken on days 0, 2 and every 4 d thereafter.

Fresh washed leaves were indoors dried at 25 ± 2 °C during 3 weeks. Seeds without tegument and callus, were oven dried at 50 °C for 48 h, dried samples were ground with a mortar and sieved through a mesh size 53 µm.

Ultrasound assisted extraction (UAE)

UAE was performed with an ultrasound bath Branson (2510R-MTH, CT, USA) with automatic control of time and temperature and ultrasound frequency of 40 kHz. 500 mg of grounded biomass dry weight (dw) were placed into a 50 mL borosilicate glass conical Erlenmeyer flask, then 20 mL of ethanol 80% (v/v) were added, and sonicated at 40 ± 5 °C during 30 min (Pandey et al., 2018; Dumitraşcu et al., 2019). During sonication flasks were suspended in the water without contact with the bottom of the ultrasonic bath, subsequently they were vortexed. Supernatant was filtered, concentrated to dryness at 25 ± 2 °C, and solubilized in 500 µL of HPLC grade MeOH (Sigma-Aldrich®) for chromatographic analysis (Saeed et al., 2006; Liu et al., 2013).

Phorbol esters (PEs) rich defatted extract

500 mg of dried sample were packed in a filter paper cartridge and defatted in a Soxhlet equipment with petroleum ether (60−80 °C) (Sigma-Aldrich®) for 4 h. Petroleum ether (Fermont®) extract was concentrated using rotary evaporator at 40 °C, 90 rpm, and 900 mbar. The methyl esters in the resulting oil, were extracted with MeOH, later filtered and concentrated to dryness at 25 ± 2 °C, then solubilized in 500 µL of HPLC grade MeOH for chromatographic analysis (Demissie & Lele, 2010).

Thin layer chromatography

Extracts were applied on normal phase silica plates (Merck Millipore, 60 F254, Germany). Chloroform-methanol (94:6 and 75:25) were used as mobile phase, reference standards were phorbol-12-myristate 13-acetate (PMA, Sigma-Aldrich®), quercetin, and vitexin (Sigma-Aldrich®), both plates were revealed with anisaldehyde (Kathiravan & Raman, 2010).

To analyze extracts obtained by sonication-ethanol 80% and Soxhlet-methanol a mobile phase consisting of chloroform-methanol (97:3) was used. The reference standard was PMA, and the plates were cerium sulfate-revealed, then observed at 366 nm, and white light. Retention factor (Rf) and color from the spots were compared with chromatographic terpenes profiles described by Reich & Schibli (2007).

MicrOTOF Q-II analysis

Electrospray ionization analysis (ESI) was performed using a micrOTOF-Q II mass spectrometer (Bruker Daltonics, Bremen, Germany) according to León-López et al. (2015). Samples were solubilized in 500 µL of HPLC grade MeOH and filtered with a syringe filter (nylon membrane, 0.45 µm, Agilent Technologies, Santa Clara, CA, USA). The molecular ions related to the extracts were analyzed in positive ion mode (ESI +). 20 µL of sample were directly injected into the evaporation chamber, capillary potential was −4.5 kV, gas temperature of 200 °C, drying gas flow of 4 L min−1 and nebulizer gas pressure of 0.4 Bar. Detection was performed at 50–3000 m/z. The predictive structures of the MS/MS partitioning profile were established utilizing the Competitive Fragmentation Modeling for Metabolite Identification (CFM-ID. Version 3.0, 2019) platform from Wishart-lab (http://cfmid3.wishartlab.com), which is referred to in the PubChem-NCBI site. Relative abundance was calculated according to Scigelova et al. (2011).

Results

Establishment of callus culture

Dedifferentiation cell was not observed in leaf blade explants. Petiole explants showed tissue dedifferentiation since seventh day of culture and complete process was evident at the day 21 (Fig. 1). Friable and light green callus was obtained on MS media added with both combinations: NAA (1.5 mg L−1), BAP (1.5 mg L−1), and NAA (3.0 mg L−1) and BAP (3.0 mg L−1).

Figure 1 Cell dedifferentiation of petiole explants from both toxic and non-toxic varieties of Jatropha curcas.

(A-D) Explants from non-toxic variety throughout dedifferentiation experiment (0, 7, 14, and 21 d, respectively), (E-H) Explants from toxic variety throughout dedifferentiation experiment (0, 7, 14, and 21 d, respectively). Both induced on MS culture medium added with NAA (1.5 mg.L-1) and BAP (1.5 mg.L-1).

Thin layer chromatography (TLC) analysis

TLC showed differences in compounds accumulation during time culture (2, 6, 10, 14, 18, 22, 26, 30, 34 and 38 d). Regard diterpenes, spots with Rf of 0.71 and 0.27 showed higher intensity along this period (Fig. 2A), unlike flavonoids in which spots with Rf of 0.77 and 0.58, decreased throughout the same culture period (Fig. 2B). These results suggest that the accumulation of diterpenes and flavonoids was inversely related during callus development. To obtain diterpenes the Soxhlet-methanol extraction was more efficient than sonication-ethanol 80%. TLC analysis of extracts obtained by both methods evidenced differences in the size and intensity of spots in regard to: extraction method, variety (toxic and non-toxic), and plant material (seeds, leaves and callus) (Fig. S1).

Figure 2 Identification of both diterpenes-type (A), and flavonoids-type (B) compounds in seeds, leaves, and callus of Jatropha curcas, through thin layer chromatography.

Lanes from 2 to 38 correspond to extracts of callus of non-toxic variety throughout 38 d of culture, NTS= Non- toxic variety-seeds, PMA= Phorbol-12-myristate-13-acetate (Sigma) reference standard (Rf 0.42), V= Vitexin (Sigma) reference standard (Rf 0.42), and Q= Quercetin (Sigma) reference standard (Rf 0.37). A) The spots intensity increased throughout to culture time (Rfs 0.71, and 0.27), mobile phase chloroform-methanol (94:6). B) The spots intensity decreased throughout to culture time (Rfs 0.77, and 0.58), mobile phase chloroform-methanol (75:25). Plates were revealed with anisaldehyde.

Figure 3 Spectrophotometrical analysis of phorbol esters in extracts of Jatropha curcas seeds.

MS/MS fragmentation profile of the molecular ion m/z 365 [M+H]+ related to 12-deoxy-16-hydroxy-phorbol, which is the structural core from Jatropha curcas-phorbol esters (referred as Jatropha factors). Predictive structures obtained through CFM-ID platform from each ionized fragment.

MicrOTOF Q-II and competitive fragmentation modeling for metabolite identification platform (CFM-ID)

Phorbol esters (PEs) analysis

Fragmentation profile analysis from seeds extract from both varieties showed several highs signals one of them with m/z of 365 [M+H]+ corresponding to 12-deoxy-16-hydroxy-phorbol (DHP), which is the fundamental structural core of the PEs. The MS/MS analysis of this molecular ion showed fragments with m/z of 295, 276, 234, 203, 185 and 127 [M+H]+ which is similar to the fragmentation profile of DHP presented in CFM-ID platform (Fig. 3), this suggests the identification of that molecular structure in all of the extracts obtained from seeds and callus of both toxic and non-toxic varieties. Based on signals intensities from 14 d and 38 d callus extracts from both varieties, it is suggested that the accumulation of DHP is time-dependent. Since, the corresponding signal was higher in callus of 38 d than in those of 14 d (Fig. 4). Furthermore, two signals with m/z of 547 and 591 [M+H]+ were observed, so it is proposed that they are related with the fragmentation profile of the signal with m/z of 711 [M+H]+ corresponding to any of the Jatropha factors (C1 or DHPB to C6) which nucleus structure is DHP (Wink et al., 2000; Haas, Sterk & Mittelbach, 2002) (Fig. S2). Table 1 shows the relative abundance of DHP molecular ion (m/z 365 [M+H]+) on 14 d and 38 d callus extracts from both varieties, evidencing the increment of these compounds through the callus development.

Flavonoids analysis

On the other hand, the main group of compounds in Jatropha leaf extracts are flavonoids, among them the apigenin, nevertheless, it is important to refer that the natural condition of flavonoids in the plants is in glycosylated form. On this regard, another of the highest signals observed at the chromatograms was the m/z of 381 [M+H]+ ion, the MS-MS experiment of this signal and the proposed structures obtained by CFM-ID platform allowed to relate that molecular ion (m/z 381 [M+H]+) to the fragmentation profiles of apigenin 6-C-α-L-arabinopyranosyl-8-C- β-D-xylopyranoside, and of apigenin 4′-O-rhamnoside (Fig. 5). The fragmentation signals and their corresponding predictive structure were also related for vitexin (m/z 433 [M+H]+), vitexin 4′-O-glucoside-2″-O-rhamnoside (m/z 741 [M+H]+), vicenin-2 (m/z 595 [M+H]+), and vicenin-2,6″-O-glucoside m/z 757 [M+H]+ (Fig. S3). Table 1 shows the relative abundance of six tentatively identified compounds by relating their molecular ions on 14 d and 38 d callus extracts from both varieties. Inversely to observed on DHP related signal (m/z 365 [M+H]+), the intensity of the molecular ion related with glycosylated apigenin (m/z 381 [M+H]+) diminished (Fig. 6).

Figure 4 Mass spectra from callus extracts of J. curcas showing the relative intensity of the molecular ion m/z 365 [M+H]+ related to the structural core of the Jatropha-phorbol esters.

Callus extracts from toxic variety: (A) 14 d of culture; (B) 38 d of culture; non-toxic variety: (C) 14 d of culture, (D) 38 d of culture. The relative intensity from molecular ion m/z 365 [M+H]+ increased throughout culture time.

Table 1 Tentative compounds identified by ESI-MS in hydroalcoholic extracts from seeds, leaves, and callus of 14 and 38 d of culture from both toxic and non-toxic Jatropha curcas L. varieties.

Compound type/ name	Elemental composition	Mass	Fragment ions in positive ion mode (m/z)	Plant material	Time of culture (d)	Variety	Relative abundance (%)	
Phorbol	
12-deoxy-16-hydroxy-phorbol	C20H28O6	364.4	127, 185, 203, 234, 276, 295				
Seeds		T	64.70	
	NT	21.05	
Callus	14	T	14.28	
NT	8.69	
38	T	30.00	
NT	25.00	
Glycosylated Flavonoids	
Apigenin 6-C- α-L-arabinopyranosyl-8-C- β-D-xylopyranoside (m/z 535 [M+H]+), and Apigenin 4′-O-rhamnoside (m/z 417 [M+H]+)	C25H26O13 and C21H20O9	534.47 and 416.4	381				
Leaves		T	45.83	
	NT	100	
Callus	14	T	100	
NT	100	
38	T	100	
NT	62.50	
Vitexin (m/z 433 [M+H]+)	C21H20O10	432.37	415				
Leaves		T	27.7	
	NT	70.00	
Callus	14	T	10.34	
NT	4.76	
38	T	11.11	
NT	<12.50	
Vitexin 4′-O-glucoside-2″-O-rhamnoside (m/z 741 [M+H]+)	C33H40O19	740.7	577				
Leaves		T	33.3	
	NT	42.84	
Callus	14	T	1.42	
NT	6.36	
38	T	12.50	
NT	8.75	
Vicenin-2 (m/z 595 [M+H]+)	C27H30O15	594.5	503				
Leaves		T	25.00	
	NT	29.16	
Callus	14	T	7.14	
NT	4.00	
38	T	6.25	
NT	3.75	
Vicenin-2,6″-O-glucoside (m/z 757 [M+H]+)	C33H40O20	756.7	757				
Leaves		T	<9.09	
	NT	8.00	
Callus	14	T	1.42	
NT	1.73	
38	T	2.22	
NT	1.25	
Notes.

T Toxic

NT Non-toxic

Discussion

The highest callus induction (95.5%) was observed in petiole explants on MS medium added with NAA (3.0 mg L−1) and BAP (3.0 mg L−1), the second best result (87.7%) was obtained with NAA (1.5 mg L−1) and BAP (1.5 mg L−1), in contrast to reported by Nassar et al. (2013), who observed dedifferentiation with NAA and BAP at 0.5 mg L−1 of each one plant growth regulator. Explants dedifferentiation reported in this work was similar to reported by Kumar et al. (2015). The follow up of the explants dedifferentiation process, every 7 d showed callus formation on explants starting on the seventh day. Dedifferentiation began at the cutting sites as expected (Sujatha, Makkar & Becker, 2005; Nogueira et al., 2011; Ovando-Medina et al., 2016). The callus obtained was light green and friable, similar to reported by Hernández et al. (2015). It has been reported that high auxins concentrations could affect production and accumulation of secondary metabolites (Kim et al., 2007), hence, according to our results, it is suggested the use of the lowest effective concentration, 1.5 mg L−1 for both growth regulators. Muñoz-Valverde et al. (2003) concluded that BAP is determinant to induce callus formation in foliar explants of J. curcas. Likewise, Suárez & Salgado (2008) reported that the presence of NAA induce callus formation in Stevia rebaudiana, and this effect could be increased when adding BAP. On the other hand, Solange et al. (2002) determined that the use of NAA and BAP in equal proportion induces callus formation from leaf explants of Tridax procumbens. Coutiño-Cortés et al. (2013) reported the callus induction in J. curcas leaf explants at 10 d of culture, and total explant-cell dedifferentiation at 20 d using 2, 4-D, BAP and KIN, while in this work, petioles dedifferentiation started at 7 d and total explant-cell dedifferentiation was achieved at 21 d. These results support that synergy between NAA and BAP is essential to achieve a high dedifferentiation degree. Stable callus culture conditions for the two varieties of Jatropha curcas were established.

The PEs are responsible for the toxicity in the plant (Devappa, Makkar & Becker, 2011; Sabandar et al., 2013; Zhang et al., 2017). There are varieties of Jatropha curcas denominated as toxic and non-toxic (Makkar et al., 1997). The non-toxic varieties have PEs concentration lower than 0.86 mg/g of seed on dry basis (He et al., 2011). Martínez-Herrera et al. (2006) detected high levels of PEs in seed oil from the municipality of Coatzacoalcos, Veracruz, México, but did not detect PEs in seeds from the municipality of Yautepec, Morelos, México. This corroborates the differences between the seeds of the two varieties used in this study.

Figure 5 Fragmentation profile (MS/MS) of the molecular ion m/z 381 [M+H]+, observed in leaves extracts, and related to fragmentation of two glycosylated apigenin (apigenin (6-C- α-L-arabinopyranosyl-8-C-β-D-xylopyranoside m/z 535 [M+H]+, apigenin 4’-O-rhamnoside m/z 417 [M+H]+)).

(A) Fragmentation profile of the molecular ion m/z 381 [M+H]+; (B) Apigenin 6-C-α-L-arabinopyranosyl-8-C-β-D-xylopyranoside m/z 535 [M+H]+, and Apigenin 4’-O-rhamnoside m/z 417 [M+H]+; (C) structures predicted to each molecular ion (381, 355, 335, and 219 m/z), obtained from CFM-ID platform.

Figure 6 Mass spectra of callus extracts from both toxic and non-toxic varieties of J. curcas at 14 and 38 d culture, showing the relative intensity of the molecular ion m/z 381 [M+H]+ related to the fragmentation profile from two glycosylated apigenin.

(A and C) Extracts of J. curcas callus from J. curcas-toxic variety (14 and 38 d, respectively). (B and D) Extracts of J. curcas callus from non-toxic variety (14 and 38 d, respectively). The relative intensity from molecular ion m/z 381 [M+H]+ diminished throughout culture time.

Regard, to TLC profile analysis, it has been reported that methanolic extraction from seed-oil facilitates separation and availability of methyl ester type compounds, mainly phorbol esters (PEs) (Demissie & Lele, 2010; Devappa, Bingham & Khanal, 2013). The detection by TLC of PEs in seed methanolic extracts from toxic and non-toxic J. curcas varieties was reported (Devappa, Makkar & Becker (2012), they reported higher spots intensity from toxic variety than from non-toxic, when plates were observed at 366 nm UV light, this result is similar to that observed in this work (Fig. S1). Makkar & Becker (2009) detected higher PEs accumulation in seeds than in leaves extracts. Similar results were obtained in this work, even with different method of extraction. Nevertheless, these results are different of that obtained by Martínez-Herrera, Chel-Guerrero & Martínez-Ayala (2004), because they reported 96% of PEs extraction through hydroalcoholic extraction, quantified by HPLC; while, in this work the intensity of the spots was higher on Soxhlet-methanol extracts than hydroalcoholic extraction (Fig. S1). Using TLC, differences between dedifferentiated cell extracts of both varieties of Jatropha curcas were evidenced.

On the other hand, Hirota et al. (2017) reported the identification of DHP as the fundamental structural core which is derived from 12-deoxy-16-hydroxy-phorbol-4′-[12′,14′-butadienyl]-6′-[16′,18′,20′-nonatrienyl]- bicycle [3.1.0] hexane-(13-O)-2′-[carboxylate]- (16-O)-3′- [8′-butenoic-10′]ate (DHPB or Jatropha factor C1), identified as DHPB-Na adduct m/z 733 [M+Na]+. Furthermore, DHPB m/z 711 [M+H]+ and DHP m/z of 365 [M+H]+ were also reported in J. curcas seeds (Wink et al., 2000). Even more so, Haas, Sterk & Mittelbach (2002) reported the identification of diterpenes named Jatropha factors C2 to C 6through ESI-MS m/z 711 [M+H]+ and of DHP (m/z of 365 [M+H] +). Furthermore, Nishshanka et al. (2016) identified six phorbol esters in J. curcas seeds by LC-MS, which have the same core (DHP), at the so named Jatropha factors (C1 to C6).

Regard to PEs identification by ESI-MS analysis, Verardo et al. (2019) identified six phorbol esters in J. curcas seeds with m/z of 711 [M+H]+, which have the same fundamental structural core (DHP) m/z of 365 [M+H]+ which is coupled to diterpenes of 24 carbon structures named Jatropha factors from C1 (DHPB) to C6. The relative intensity of the molecular ion m/z 365 [M+H] + was higher in seeds extracts from toxic variety, than in seed extracts from non-toxic variety (Data not shown). While in callus, the relative intensity is higher in toxic and non-toxic varieties callus of 38 d of culture (Figs. 4B, and 4D, than in toxic and non-toxic varieties callus of 14 d of culture (Figs. 4A, and 4C). These results could suggest the presence of PEs coupled to DHP in the samples analyzed and that their accumulation is differential in regard to the variety-derived cell culture and throughout the time of culture. These results suggest that their accumulation of DHP is time dependent. This ESI-MS analysis allowed to corroborate the results obtained by TLC (Fig. 2A). Nevertheless, the relative intensity of the signals observed in extracts from callus were lower than that obtained from seeds extracts as reported by Demissie & Lele (2010). By ESI-MS, differences in the relative intensity of the signal corresponding to DHP were observed between the callus extracts of both varieties, being higher in the toxic variety in addition to that in calluses at 38 d of culture, it was higher than in the 14 d.

By other hand, phenolic compounds are ubiquitously produced by plants (Kumar & Goel, 2019), the main role of phenols in plants is to protect them from biotic or abiotic stress (Pereira, 2016). These properties include antimicrobial, insecticidal, antiparasitic, antiviral, anti-ulcerogeBynic, cytotoxic, antioxidant, anti-hepatotoxic, anti-hypertensive and anti-inflammatory activities (Oskoueian et al., 2011; Papalia, Barreca & Panuccio, 2017). Flavonoids are recognized as polyphenols. Several of them have been identified in Jatropha genus, such as apigenin glycosides, vitexin, and isovitexin which have been considered as chemiotaxonomic compounds from the genus (Abdelgadir &Van Staden, 2013; Huang et al., 2014).

The tentative identification of glycosylates-flavonoids through microQTOF-QII has been already reported (Pezzini et al., 2019). In this regard Xie et al. (2003) reported the apigenin 6-C- α-L-arabinopyranosyl-8-C- β-D-xylopyranoside m/z 535 [M+H]+. Likewise, this result may be related to that obtained by Abd-Alla et al. (2009) who identified apigenin and its aglycone as majoritarian flavonoids in J. curcas leaves, as well as, that obtained by Reena, Nand & Sharma (2008) who reported to apigenin as major flavonoid in the same species. Those reports differ from that published by Papalia, Barreca & Panuccio (2017) who identified to vitexin and vicenin-2 as the majoritarian flavonoids.

The results obtained by microQTOF-QII of the molecular ion m/z 381 [M+H]+ through the MS/MS experiment, and the predictive structures obtained through the CFM-ID platform allowed to suggest the relation of the structures from the molecular ion m/z 381 [M+H]+ with the fragmentation profile from apigenin 6-C- α-L-arabinopyranosyl-8-C- β-D-xylopyranoside m/z 535 [M+H]+, which was identified through ESI-MS in Viola yedoensis (Xie et al., 2003) and apigenin 4′-O-rhamnoside m/z 417 [M+H] +, which was identified in Olea europaea (Pieroni et al., 1996).

Based on the molecular ion, MS-MS fragmentation profile and the predictive structures obtained by CFM-ID platform, it is suggested the tentative identification of vitexin m/z of 433 [M+H]+, vicenin-2 m/z of 595 [M+H]+, and vitexin 4′-O-glucoside-2″-O-rhamnoside m/z of 741 [M+H]+ in leaves and callus from both varieties. These results are similar to obtained by Huang et al. (2014) who identified vitexin m/z of 433 [M+H]+ in J. curcas leaves. This flavonoid was also identified by ESI-MS in Parkinsonea aculeata m/z of 431 [M-H]− (Hassan, Abdelaziz & Al Yousef, 2019). In this work it is also suggested the tentative identification of vicenin-2,6″-O-glucoside m/z 757 [M+H]+ which has not been reported to Jatropha curcas, but to Stell aria holostea (Bouillant et al. 1984) (Fig. S2). By the MS-MS fragmentation profile, the identification of six glycosylated flavonoids is suggested, it was observed that relative intensities signals related to flavonoid related molecular ion m/z 381 [M+H]+ in callus of 14 d was higher than callus of 38 d, differences that were not observed between calluses of the different varieties.

Conclusions

Stable dedifferentiated cells culture from petiole explants of Jathopha curcas, were stablished from toxic and non-toxic varieties on MS medium added with NAA and BAP. Thin layer chromatography and mass spectrometry, suggest an inverse relationship between phorbol esters and flavonoids accumulation in callus throughout the time of culture. The tentative identification of diterpene type compounds such as 12-deoxy-16-hydroxy-phorbol and Jatropha factors by ESI-MS in seed and callus (14 and 38 d), as well as, the presence of six flavonoids glycosides in leaf and callus, in extracts from both toxic and non-toxic varieties of J. curcas, is suggested. Both of them, groups of compounds reported with bioactive activity with pharmaceutic/agroindistrial potential.

Supplemental Information

Supplemental Information 1 Thin layer chromatogram of extracts obtained from seeds, leaves, and callus of two J. curcas-varieties with two extraction methods for the identification of phorbol esters (Rf’s 0.81, 0.53, and 0.38)

The extracts obtained with ethanol 80% - sonication are referred with numbers (1 8). The extracts obtained with Soxhlet methanol are referred with letters (A H). PMA: Phorbol-12-myristate-13-acetate Rf 0.22 (Sigma, PE reference standard). Toxic variety seed (1 and A), Non-toxic variety seed (2 and B), Toxic variety leaves (3 and C), Non-toxic variety leaves (4 and D), Toxic variety-callus 14 d (5 and E), Toxic variety-callus 38 d (6 and F), Non-toxic variety-callus 14 d (7 and G), Non-toxic variety-callus 38 d (8 and H). Mobile phase chloroform-methanol (97:3), cerium sulfate-revealed, observed at 366 nm UV light.

Click here for additional data file.

Supplemental Information 2 Mass spectra of callus extracts of J. curcas showing the relative abundance of the molecular ions related to structural core of the Jatropha-phorbol esters and flavonoid-type compounds

Toxic variety-callus 14 d extract (A), toxic variety-callus 38 d extract (B), non-toxic variety-callus 14 d extract (C), and non-toxic variety-callus 38 d extract (D).

Click here for additional data file.

Supplemental Information 3 Predicted structures related with the fragmentation profile from six flavonoids identified through ESI-MS from calluses extracts of non-toxic Jatropha curcas

It is included the predictive structure corresponding to vicenin-2,6”-O-Glucoside m/z 757 [M+H]+ which is not reported to, but it is to Stellaria holostea.

Click here for additional data file.

We recognize Centro de Nanociencias y Micro y Nanotecnologías (CNMN-IPN) for the experimental service with the micrOTOF-Q II spectrometer. IPN). We thank Dra. Silvia Evangelista Lozano from Centro de Desarrollo de Productos Bióticos (CEPROBI-IPN), for providing the non-toxic variety plant material, thanks also to Dr. Edwin Javier Barrios Gómez from Instituto Nacional de Investigaciones Forestales, Agrícolas y Pecuarias (INIFAP-Mor, Mexico), for facilitating the toxic variety plant material.

Additional Information and Declarations

Competing Interests

Author Contributions

Data Availability

The authors declare there are no competing interests.

Gerardo Leyva-Padrón performed the experiments, prepared figures and/or tables, and approved the final draft.

Pablo Emilio Vanegas-Espinoza performed the experiments, analyzed the data, authored or reviewed drafts of the paper, and approved the final draft.

Alma Angélica Del Villar-Martínez conceived and designed the experiments, analyzed the data, authored or reviewed drafts of the paper, and approved the final draft.

Crescencio Bazaldua conceived and designed the experiments, analyzed the data, prepared figures and/or tables, authored or reviewed drafts of the paper, and approved the final draft.

The following information was supplied regarding data availability:

The raw data is available in the Supplemental Files.

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
