# Peer review of "Chemical analysis of callus extracts from toxic and non-toxic varieties of Jatropha curcas L"

_PeerJ, doi:10.7717/peerj.10172_

## Round 0.1 · original submission · Major Revisions

The authors are requested to carefully address all the comments made by the reviewers.

Reviewer 1 ·

Basic reporting

This is an interesting work and adds some scientific information to this important medicinal plant species. However, authors need to connect the story as the whole MS is missing the flow. The English language is Ok. The chosen title is not the best one.

Experimental design

Jatropha is a widely studied plant and has important commercial value. I suggest enriching the introduction and discussion part on the basis of current findings. What is the originality of your work? Why have you chosen toxic and non-toxic varieties? are there any similarities or differences in the chemical profiles on your varieties and other varieties reported by other authors?

Validity of the findings

I think the conclusion past can be improved. It would be nice to summarize all findings in the chemical profile of different parts and varieties in a table. This will help the readers to understand the findings quickly. Some figures have poor quality (i.e., fig 6, fig 4). As a reader, I could not find the importance of this work. Please highlight and specify your findings. Why did you plant this work? do you have worthy findings?

Reviewer 2 ·

Basic reporting

1.1- English is correct.
1.2- Sufficient and relevant prior literature is cited
1.3- Quality of Figures and Table is sufficient
1.4- Results are in part relevant to the hypothesis (see validity of the findings)

Experimental design

2.1- This work fall into the aims and scope of the journal
2.2- Research question is well defined
2.3- Methods have to be improved for publication: quantification of the different compounds (at least relative) is strongly needed to support the conclusions
2.4- Methods section has to be more clearly described. In particular, US apparatus as well as US extraction parameters are not described in details. The choice of the extraction parameters is not justified. The description of the US apparatus did respect the conventional description. The authors will find these standards in the paper “Ultrasonically assisted extraction (UAE) of natural products some guidelines for good practice and reporting” published in Ultrasonics Sonochemistry, 25 (2015) 94-95.
The reference cited by the auhtors is not related (ie, on podophyllotoxin extraction from another plant source!)

Validity of the findings

3.1- impact and novelty are basic
3.2- no quantification therefore no statistical analysis, this is a drawback for the present work.
3.3- conclusions have to be rewritten to answer the main questions of the present work

Additional comments

1. The need of in vitro culture initiation is not described by the auhtors
2. Quantification (at least relative) is vital to present firm conclusion
3. Title is not correct. Callus are not stabilized so "during callus initiation" should be used instead
4. Difference between toxic and non-toxic is unclear in the conclusion (again quantification and statistical analysis are required)
5. Objectives of the present work are not clearly exposed

·

Basic reporting

The authors do not refer to the occurrence of J. curcas in other countries to compare their toxicity. In these aspects, the author should introduce some aspects broader out of the local place studying. Its relevance should be mentioned because of the chemotype variety is not discussed to justify the work. The article should include sufficient information if or not chemotypes are one of the important aspects of the presence or absence of the aim metabolite in the introduction, and relate it to J. curcas toxicity.

Figure 2, it is not so clear, it must be sufficient clearness to show the reference standard. When the extract amount is applied as spot the band appears as spots when applied as a horizontal band (like on electrophoresed gel), it should appear in this paper in the horizontal band form, the results present in this article are not so conventional. It needs an explanation.

Besides, it is important to explain the TLC variation in the volume application to justify the not uniform profiles amounts.
Inline 208, the statement is not appropriate, because TLC does not identify, It does detect the aim metabolite, groups or class. it is important to review.

Experimental design

No comment

Validity of the findings

No comment

·

Basic reporting

1-Fair
2-Fair
3-Fair
4-Fair

Experimental design

1-Fair
2-Fair
3-Fair
4-Fair

Validity of the findings

1-Fair
2-Fair
3-Fair
4-Fair

Additional comments

The information provided in this article may aid in enhancing awareness of Jatropha plants and its products with respect to the potential toxicity of Jatropha.
1- Generally, the paper needs English editing for some spelling and grammar mistakes along with many unclear sentences.
2- What is the main clear idea(s) we get from this paper? Please clarify in conclusion section.
3- Add photo for seeds and leaves of toxic and non toxic Jatropha varieties at the initial stage of cultivation.
4- About 61 references include only five references raised last five years, please update with recent references.
5- You are kindly requested to follow the corrections made within attachment file.

---

## Round 0.2 · accepted · Accept

The manuscript has been improved. The reviewers have suggested that it can be accepted for publication now.

Reviewer 1 ·

Basic reporting

English is acceptable
References have been updated
The figure and table have been modified accordingly

Experimental design

All the questions and suggestions have been incorporated in the revised version by the authors

Validity of the findings

All the questions and suggestions have been incorporated in the revised version by the authors

Reviewer 2 ·

Basic reporting

see General comments for the author

Experimental design

see General comments for the author

Validity of the findings

see General comments for the author

Additional comments

The Authors have perfectly answered to all my queries.
From my point of view, this work is now acceptable for publication.